# Imination of Microporous Chitosan Fibers—A Route to Biomaterials with “On Demand” Antimicrobial Activity and Biodegradation for Wound Dressings

**DOI:** 10.3390/pharmaceutics14010117

**Published:** 2022-01-04

**Authors:** Alexandru Anisiei, Irina Rosca, Andreea-Isabela Sandu, Adrian Bele, Xinjian Cheng, Luminita Marin

**Affiliations:** 1“Petru Poni” Institute of Macromolecular Chemistry, Gr. Ghica Voda Alley, 41A, 700487 Iasi, Romania; anisiei.alexandru@icmpp.ro (A.A.); rosca.irina@icmpp.ro (I.R.); sandu.isabela@icmpp.ro (A.-I.S.); bele.adrian@icmpp.ro (A.B.); 2School of Chemistry and Environmental Engineering, Wuhan Institute of Technology, Wuhan 430073, China; chxj606@163.com

**Keywords:** chitosan, microporous nanofibers, imine linkage, wound healing, controlled biodegradation, antimicrobial activity, biocompatibility, bioadhesivity

## Abstract

Microporous chitosan nanofibers functionalized with different amounts of an antimicrobial agent via imine linkage were prepared by a three-step procedure including the electrospinning of a chitosan/PEO blend, PEO removal and acid condensation reaction in a heterogeneous system with 2-formylphenylboronic acid. The fibers’ characterization was undertaken keeping in mind their application to wound healing. Thus, by FTIR and ^1^H-NMR spectroscopy, it was confirmed the successful imination of the fibers and the conversion degree of the amine groups of chitosan into imine units. The fiber morphology in terms of fiber diameter, crystallinity, inter- and intra-fiber porosity and strength of intermolecular forces was investigated using scanning electron microscopy, polarized light microscopy, water vapor sorption and thermogravimetric analysis. The swelling ability was estimated in water and phosphate buffer by calculating the mass equilibrium swelling. The fiber biodegradation was explored in five media of different pH, corresponding to different stages of wound healing and the antimicrobial activity against the opportunistic pathogens inflicting wound infection was investigated according to standard tests. The biocompatibility and bioadhesivity were studied on normal human dermal fibroblast cells by direct contact procedure. The dynamic character of the imine linkage of the functionalized fibers was monitored by UV-vis spectroscopy. The results showed that the functionalization of the chitosan microporous nanofibers with antimicrobial agents via imine linkage is a great route towards bio-absorbable wound dressings with “on demand” antimicrobial properties and biodegradation rate matching the healing stages.

## 1. Introduction

Wound healing is a frequent skin trauma that severely impacts public health. As an example, every day over 30,000 people worldwide suffer severe burns which require medical attention [1]. Among them, invasive infection is responsible for 51% of burn deaths, especially in low- and middle-income countries with a deficient care system [2].

This social and economic context guided considerable efforts toward the development of efficient approaches for wound healing. Many materials were proposed to this aim, such as films, foams, hydrocolloids, hydrogels, hydrofibers or sprays [3,4,5,6]. To fulfill the main requirements for an efficient wound dressing, natural and naturally derived polymers are ideal candidates due to their biocompatibility and lack of toxicity. Chitosan biopolymer received special attention regarding this aim, due to its beneficial properties for wound healing: it is biocompatible, bioadhesive, non-toxic, non-allergenic, hemostatic and antimicrobial, and promotes the growth of granulation tissues. Chitosan-based hydrogels, membranes, sponges or fibers, were investigated as wound dressing candidates and they showed remarkable results [7,8,9,10,11,12]. The main shortcoming of these materials appeared to be the adhesion to the wound surface requiring mechanical debridement, which affects the new epithelial tissue while it is highly traumatizing for the patient. A pathway to solving this issue appears to be the development of bioabsorbable materials, and a few studies that considered water-soluble polymers such as gelatin [13] or carboxymethyl [14] showed promising findings.

Among the chitosan-based materials, the nanofibers promise high potential for wound dressing, as they have good morphological similarities to the natural extracellular matrix, good absorbency, semi permeability and excellent conformability. Besides this, the nanofibers can be prepared by electrospinning—a low-cost green technique [15]. On the other hand, due to the limited solubility of chitosan, neat chitosan nanofibers are difficult to obtain by electrospinning. This limitation restricted the investigation of neat chitosan nanofibers for wound healing application, blends with synthetic high molecular weight polymers, such as polyethylene glycol and poly (vinyl alcohol) being usually obtained [16]. While biocompatible, these synthetic polymers used as co-spinning agents are not in vivo biodegradable, hindering the preparation of bioabsorbable materials [17,18].

In the light of these data, we designed new multifunctional biomaterials targeted at fulfilling the main requirements of an ideal wound dressing. Thus, we prepared microporous neat chitosan fibers (pore size up to 2 nm) by electrospinning and further functionalized them with an antimicrobial agent via imination. The microporous nature of the fibers was hypothesized to improve the gas exchange and exudate drainage; the antimicrobial agent should be slowly released “on demand” due to the imination reversibility, and the biodegradation of chitosan in lysozyme medium of wounds should assure the circumventing of traumatic debridement. Besides this, the biodegradation products of chitosan, which are natural metabolites, should have a positive effect on the healing process [19]. Moreover, the electrospinning technique used for fibers’ preparation is a green, low-cost method, counting for a low price of the final materials [15].

## 2. Experimental

### 2.1. Materials

Chitosan (126 kDa, DA = 97%) was prepared by basic hydrolysis (Appendix A). 2-Formylphenylboronic acid (Aldrich, 97%) was purified by column chromatography (Appendix A). Polyethylene oxide (1000 kDa), acetic acid (99, 89%), sodium hydroxide (95%) and ethanol (98.89%) were purchased from Aldrich. Ethanol was dried on molecular sieves before use.

### 2.2. Fiber Preparation

The imino-chitosan fibers were prepared following three steps:(1)Chitosan/poly (ethylene oxide) (**CS/PEO**) fibers were prepared by electrospinning a 2.1 g/mL solution of **CS/PEO** (2/1, *w*/*w*) in 80% acetic acid. The solution was loaded in a 5 mL syringe with a blunt needle with an inner diameter of 0.8 mm. The electrospinning was done at room temperature, applying a voltage of 7 kV, a tip-to-collector distance of 10 cm, a flow rate of 0.4 mL/h and a rotary drum collector speed of 800 rpm.(2)Microporous neat **CS** fibers were prepared by washing the PEO [20] from **CS/PEO** fibers with a 5% NaOH solution in order to remove the residual acetic acid, and then with distilled water to remove the PEO and to reach the neutral pH. Finally, the wet fibers were lyophilized in order to preserve the porosity gained by PEO washing.(3)The CS fibers were functionalized by imination reaction with 2-formylphenylboronic acid in a heterogeneous system, to give imino-chitosan fibers, coded **BC**. The fiber mat was immersed into a vessel containing 10 mL solution of aldehyde in ethanol and kept sealed at 55 °C, for 24 h. To obtain a series of **BC** fibers with different content of imine units, the molar ratio between glucosamine units of chitosan and 2-formylphenylboronic acid was varied from 1/1 up to 6/1 (Figure 1). When the reaction time ended, the vessel was allowed to reach room temperature and then it was unsealed to allow the ethanol removal and fibers’ drying. Then, the fibers were washed with dry ethanol to remove the unreacted aldehyde and dried in atmospheric conditions. The sample with a 1/1 molar ratio of the functional groups was also prepared and rapidly dried at 50 °C after the reaction time passed. A representation of the preparation procedure and the sample codes is illustrated in Figure 1.

### 2.3. Equipment and Measurements

The fibers were frozen in liquid nitrogen and then lyophilized using Labconco FreeZone Freeze Dry System equipment (Kansas, MO, USA) for 24 h at −54 °C and 1.514 mbar.

FTIR spectra were recorded on an FT-IR Bruker Spectrophotometer VERTEX 70 (Billerica, MA, USA), using potassium bromide tablets as support for the milled fibers samples. The spectra were recorded from 4000 to 600 cm^−1^, with 32 scans and a resolution of 4 cm^−1^, and they were further processed with OPUS 6.5 software.

Proton nuclear magnetic resonance (^1^H-NMR) spectra were recorded using a Bruker Avance DRX 400 MHz spectrometer (Billerica, MA, USA), the chemical shifts were displayed in parts per million. In order to quantify the imination degree, the fibers were dissolved in 2% AcOH in D_2_O to assure the complete shifting of the imination equilibrium to the reagents. The conversion degree of the amine groups into imine units was calculated using the following equation.
η = (A_CH=O_)/(A_H__2_ ∗ 0.97) ∗ 100 (1)
where A_CH=O_ is the integral of the chemical shift of the aldehyde protons; A_H__2_ is the integral of the chemical shift of the H2 of chitosan and 0.97 is the deacetylation degree established by ^1^H-NMR spectroscopy (Appendix A) [21]. The spectra were measured on three different samples and the results were presented as the average value.

Thermogravimetric analysis was performed with a TA Instruments TGA 5500 thermogravimetric analyzer using as probe support a 100 µL platinum pan, with a heating rate of 10 °C/min, from 29 °C to 600 °C.

Polarized optical microscopy images were acquired with a Zeiss Axio Imager M2 microscope (Wetzlar, Germany).

The fibers morphology was monitored with a field emission Scanning Electron Microscope SEM EDAX-Quanta 200 (Waltham, MA, USA), operated at an acceleration voltage of 20 keV. The average diameter of the fibers was measured using the Image J software (ImageJ bundled with 64-bit Java 1.8.0_172).

The swelling was investigated by immersing the fibers into ultrapure water or PBS of different pH, and sample weighting since the weight value remained constant. The swelling degree was evaluated by calculating mass equilibrium swelling (MES) with the equation:MES = (Mf − Mi)/Mi (2)
where Mi is the initial mass and Mf is the final mass of fibers.

The in vitro enzymatic degradation was monitored in a solution of lysozyme (376 U/mL) in PBS of different pH values: 5.5, 7.4, 8.5, 9 and 10, by calculating their mass loss. The fiber biodegradation in a solution of 4830 U/mL lysozyme dissolved in PBS of pH = 7.4 was investigated too. Discs of fiber mats of 1 cm diameter, weighing 1.5 ± 0.05 mg each were immersed into 10 mL lysozyme solution and kept at 37 °C. At certain moments, the samples were removed, washed three times with distilled water in order to remove the PBS salts and subjected to gravimetric measurements and SEM imaging. The mass loss was calculated using the equation:W_loss_ = [(W_0_ − W_t_)/W_0_] ∗ 100 (3)
where W_0_ is the initial weight and W_t_ is the weight at a certain moment of the fibers.

Water vapor sorption capacity of the samples was measured in a dynamic regime by using a fully automated gravimetric analyzer IGAsorp by Hiden Analytical, following a standard protocol [22]. The weight changes that appeared by modification of humidity in the sample chamber at constant temperature were measured with an ultrasensitive microbalance integrated into the device. Sorption of water vapors was measured in a dynamic regime at 25 °C in the relative humidity (RH) range 0–90%, with a vapor pressure increase in 10% humidity steps. For desorption, 10% humidity steps were used for the entire humidity range. Before the desorption measurements, a drying process at 25 °C was realized, in flowing nitrogen (250 mL/min) until the weight of the sample was at equilibrium. The maximum water uptake was calculated with the equation:(4)$$W=\left[\left(W_{w}-W_{d}\right)/\mathrm{Wd}\right]\times 100$$
where $W_{d}$ was the weight of the dried sample and $W_{w}$ was the weight of the weighted sample. The average pore size was calculated with the equation:(5)$$\mathrm{average}\;\mathrm{pore}\;\mathrm{size}=\frac{2\times W}{100\times \rho \times A}$$
where ρ = adsorbent phase density and A is the Brunauer–Emmett–Teller (BET) surface area.

The dynamic character of the imine linkage was investigated by UV-vis spectroscopy on a Carl Zeiss Jena SPECORD M42 spectrophotometer (Wetzlar, Germany), following three scenarios. Imino-chitosan fiber samples weighting 1.5 ± 0.05 mg were immersed into a vial containing 10 mL PBS (pH = 7.4). In the first scenario, 2 mL supernatants were withdrawn at certain times: 1, 6, 24 h and replenished with fresh buffer. In the second scenario, 2 mL supernatants were withdrawn after 24 h. In the third scenario, the fiber sample was placed on a filter paper moisture in PBS (mimicking the skin) and at certain moments: 10, 20, 30 min and 1, 2, 3, 4, 24, 25, 26 h, the sample was removed and placed on another filter paper. At the end of the experiments, the filter papers were extracted in 2 mL PBS. The extracted supernatant samples were analyzed using UV-vis spectroscopy measuring the absorbance of the 2-formylphenylboric acid at 288 nm.

The antimicrobial activity was determined by the disk diffusion assay against representative bacterial and fungal reference strains: *Staphylococcus aureus* ATCC 25923, *Escherichia coli* ATCC 25922, *Candida albicans* ATCC 10231 and *Aspergillus brasiliensis* ATCC 9642. All microorganisms were stored at −80 °C in 20% glycerol. The bacterial strains were refreshed in trypticase soy broth (TSB) at 36 ± 1 °C, the yeast strain (*C. albicans*) was refreshed on Sabouraud dextrose broth (SDB) at 36 ± 1 °C and the fungal strain (*A. brasiliensis*) was refreshed in potato dextrose broth (PDB) at 25 ± 1 °C. Microbial suspensions were prepared with these cultures in sterile solution to obtain turbidity optically comparable to 0.5 McFarland standards. Volumes of 0.3 mL from each inoculum were spread onto TSA/SDA/PDA plates and the sterilized fiber samples were added to the inoculated plates. The fiber samples were prepared by cutting discs of 1 cm diameter, weighing 1.5 ± 0.05 mg each, and sterilized by 15 min exposure with UV-light (253.7 nm) on each side. The antimicrobial activity was assessed by measuring the growth inhibition under standard conditions after 24 h of incubation at 36 ± 1 °C for the bacterial strains and after 120 h of incubations at 25 ± 1 °C for *A. brasiliensis*. The tests were carried out on fiber samples of 1 cm diameter, weighing 1.5 ± 0.05 mg each, in triplicate. After incubation, the diameter of inhibition zones was measured by using Image J software (University of Wisconsin, Madison, WI, USA). All data were expressed as the mean ± standard deviation (SD) by using XLSTAT software [23]. After incubation, the remnant fiber samples were inactivated and analyzed for surface and structure modifications.

Biocompatibility test was assessed by MTS assay using the CellTiter 96^®^ AQueous One Solution Cell Proliferation Assay (Promega, Madison, WI, USA), according to the manufacturer instructions and direct contact procedure adapted from ISO 10993-5:2009 (E), with several modifications, as described by Antunes dos Santos et al. [24]. Briefly, prior to cell culture, fiber samples were cut into 7 mm diameter discs (weighing 9 ± 0.03 mg each) and sterilized by 15 min exposure with UV-light (253.7 nm). Normal fibroblasts were seeded onto sterilized nanofibers at a density of 1 × 10^5^ cells/mL into a 24-well tissue culture-treated plate, in a 1 mL culture medium/well. Cells were then incubated for 24 h. Control cells were incubated only with a culture medium. The next day, the medium was removed, and fresh medium and MTS reagent was added 1–3 h prior to absorbance readings. After the formazan formation, a triplicate of each sample was transferred into a 96-well plate and the final reading was performed at 490 nm on a FLUOstar^®^ Omega microplate reader (BMG LABTECH, Ortenberg, Germany). The tests were done in triplicate. Cell viability was expressed as a percentage of control cells’ viability. Data analysis was performed with GraphPad Prism software version 7.00 for Windows (GraphPad Software, San Diego, CA, USA). Graphical data were expressed as mean ± standard error of the mean (S.E.M.). To confirm the MTS assay, the morphology of cells exposed to the fiber mats was analyzed by optical microscopy. Brightfield images were acquired with a Leica DMI 3000B inverted microscope (Leica Microsystems, Wetzlar, Germany). In order to study the cells’ attachment to the nanofiber scaffolds, scanning electron microscopy (SEM) was used to investigate the morphology (Waltham, MA, USA) of attached cells on nanofiber dressings.

In order to study the cells’ attachment to the nanofiber scaffolds, the samples were incubated under the same conditions described above. After 24 h, the samples were removed from the 24-well plate and washed two times with PBS and fixed with 2.5% glutaraldehyde solution for 6 h [24]. Then, the samples were washed two times with ultrapure water and gradually dehydrated with increasing concentrations of ethanol (10%, 30%, 50%, 70%, 100%) for 15 min. Scanning electron microscopy (SEM) was used to investigate the morphology of attached cells on nanofiber dressings.

All experiments were done in triplicate and the data were presented as average values ± standard deviations.

## 3. Results and Discussion

A series of imino-chitosan nanofibers with different content of imine units was obtained by the condensation reaction in a heterogeneous system of **CS** nanofibers (solid phase) with 2-formylphenylboronic acid in ethanol (liquid phase), in various molar ratios of the **CS** glucosamine units and aldehyde (Figure 1, Figure 1). The neat **CS** nanofibers were prepared by **CS/PEO** electrospinning followed by PEO removal. The fibers formed a mat which could be easily manipulated, resisting well under mechanical stress (Appendix A) and presenting good adhesivity to different materials such as filter paper swollen in PBS buffer, glass, skin (Appendix A), indicating their suitability for use in wound healing.

### 3.1. Structural Characterization of the Imino-Chitosan Fibers

The imination of **CS** fibers was firstly assessed by FTIR spectroscopy (Figure 1a and Appendix A). Neat chitosan fibers showed the absorption band characteristic of the residual *N*-acetyl groups (C=O stretching) at 1645 cm^−1^ as a shoulder of the N-H bending band of the primary amine groups (1543 cm^−1^). By comparison, the imino-chitosan fibers showed a new absorption maximum at 1624 cm^−1^, characteristic of the stretching vibration of imine bonds [25]. Absorption bands characteristic to the aldehyde residue, such as in plane skeletal vibration of the C=C bonds (1564 cm^−1^) and out-of-plane bending of the B-OH (760 cm^−1^) were also present [25,26]. These spectral modifications indicated that part of the amine units of chitosan reacted with 2-formylphenilboronic acid, forming imine bonds. The bands characteristic to the newly formed products increased in intensity along the aldehyde amount used in the imination step, according to a superior conversion of amine groups into imine bonds. The band specific to the carbonyl stretching vibration of 2-formylphenylboronic acid (1730 cm^−1^) was not noticed in the spectra, indicating no free aldehyde into the functionalized fiber samples.

Another significant modification in the FTIR spectra occurred in the 3700–3000 cm^−1^ domain, characteristic for the N-H and O-H stretching, and their inter- and intra-molecular hydrogen bonds as well [25,27]. The broad band from 3242 to 3681 cm^−1^ with an absorption maximum at 3498 cm^−1^ recorded for the **CS** fibers, became wider, from 2986 to 3674 cm^−1^, and its maximum shifted to 3458 cm^−1^ for the imino-chitosan samples. This pointed out the modifications of the H-bond network along the imination reaction, possibly including the formation of new intra-molecular H-bonds between the labile hydrogen atoms of boric acid residue and the electron-rich nitrogen of the imine unit, forming an imino-boronate unit [25,28].

^1^H-NMR spectroscopy was used as a complementary characterization method, in order to confirm the imination reaction and, also, to gain a quantitative insight on the conversion degree of amine groups of chitosan into imine units. The ^1^H-NMR spectra of samples dissolved in 1% acetic acid in deuterated water showed the chemical shifts characteristic to the aldehyde and imine protons (Figure 2a). Based on the absence of characteristic aldehyde bands in FTIR spectra, their concomitant presence in ^1^H-NMR spectra was attributed to the imination reversibility in the acidic medium [17,29,30]. Increasing the acetic acid concentration to 2%, the chemical shift of the imine proton completely disappeared, and the spectra showed the chemical shifts characteristic to 2-formylphenylboronic acid and chitosan protons (Figure 1b and Appendix A). Applying Equation (1), the conversion degree of imino-chitosan fibers gave values from 14.75 to 52.87 (Figure 2b). As expected, the conversion degree increased as the amine/aldehyde molar ratio increased. Interestingly, high conversion differences were noticed between the **BC1** and **BC1R** samples (52.87% versus 14.75%) for which the only difference in the synthetic procedure consisted of the solvent removal rate (slow solvent removal at atmospheric conditions versus fast removal at 55 °C). This suggested that prolonged contact between reagents while the reaction medium became more concentrated is favorable for higher reaction yield.

The dynamic character of the imine bonds was investigated by UV-vis spectra recorded for the supernatant samples extracted at different moments in PBS in which the functionalized fibers were immersed. The presence of the absorbance characteristic to the 2-formylphenylboronic acid demonstrated the reversibility of the imine bonds (Appendix A). Moreover, the almost constant absorbance values, no matter what the time of the supernatant extraction was, demonstrated that the imination equilibrium shifted to the reagents once the aldehyde was removed from the system. Furthermore, it was observed that the rate of aldehyde release from fibers was dependent on the aldehyde removal from the media: it was fast when the aldehyde had been removed and slowed down when the aldehyde was not removed (Figure 3). It could be concluded that the consumption of the released aldehyde triggers the release of new aldehyde amounts from fibers by shifting the imination equilibrium.

### 3.2. Fiber Mat Morphology

The morphology of imino-chitosan fibers was assessed by SEM and POM microscopy and vapor sorption experiments. SEM analysis revealed that smooth nanofibers, with no beads and a mean diameter of 137 ± 21 nm were obtained by **CS/PEO** electrospinning, (Figure 4a). The mean diameter of neat chitosan fibers obtained by PEO removal increased to 176 ± 33 nm. This can be attributed to the fiber swelling during PEO removal, and preservation of the swollen morphology by lyophilization (Figure 4b).

By imination, the fiber mats became less homogeneous with an average mean diameter from 139 ± 14 to 185 ± 28 nm, and inter-fiber pores around 1 µm (Figure 4c–g). The lower diameter of the imino-chitosan fibers compared to the neat chitosan ones was attributed to a collapse of the intra-fiber pores during the drying process in atmospheric conditions. The imino-chitosan fibers appeared to stick each other, in agreement with the formation of imino-boronate units and their self-assembling leading to a crosslinking effect [20,25]. The **BC1R** sample showed the most inhomogeneous morphology and the thickest diameter, according to a preponderant imination at the fiber surface during the fast solvent evaporation.

Under polarized light, the fibers revealed a continuous birefringent texture, indicating the alignment of the chitosan macromolecules during the electrospinning leading to a crystallinity degree, which was not altered by the imination process (Figure 4h and Appendix A) [31].

Considering the fibers’ preparation procedure and SEM morphology, it was hypothesized that PEO removal created pores inside the **CS** fibers (intra-fiber pores). To analyze this hypothesis, water vapor sorption experiments were conducted.

The isotherms of all samples matched a pseudo-type II isotherm with hysteresis loop of type H3 (Figure 5a), which is usually associated with the metastability of adsorbed multilayer and delayed capillary condensation prompted by a low degree of pore curvature and non-rigid nature of the adsorbent [32]. This fits very well onto the anticipated intra-fiber morphology, as the PEO removal should lead to long, thin channels. The isotherm profile was almost similar for all samples, in agreement with a similar affinity towards water vapors, attributed to the abundant hydrophilic functionalities (-OH, -N=CH, -B (OH)_2_, -NH_2_). Slight differences in their shape were ascribed to the different proportions of functional groups, given by the different conversion degrees of amine groups into imine units. The kinetics of water vapor sorption uptake (Appendix A) showed a decreasing trend as the imination degree increased, reflected in the diminishing of the maximum water uptake from 28 to 14.7% (Figure 5b). This indicated the porosity reducing, confirming that imination took place inside the fiber pores as well. The pore size calculated with Equation (5) gave values less than 2 nm, which, according to IUPAC terminology was attributed to a microporous fiber morphology [32].

Thermogravimetric analysis can reveal information regarding the chemical composition of nanofibers and the strength of intermolecular forces [33]. In this view, the fiber’ thermograms were recorded in order to monitor (i) the PEO removal and (ii) the influence of imination on the fiber’ morphology (Figure 6 and Appendix A). All samples displayed a mass loss of 5–10%, up to 100 °C attributed to the water loss. Interestingly, the imino-chitosan fibers showed a higher water loss compared to neat chitosan fibers (10% versus 5%) suggesting a hydrophilicity enhancement by imination, a fact also observed for other imino-chitosan derivatives [34]. The **CS/PEO** fibers showed two main degradation stages, with the decomposition maxima at 259 °C and 375 °C, corresponding to the degradation of the two component polymers, chitosan and PEO [35]. In the thermogram of **CS** fibers, the decomposition stage of PEO completely disappeared confirming its successful removal, while the maximum of chitosan decomposition shifted to a higher temperature, at 297 °C. This is attributable to the stronger forces among the chitosan macromolecules, compared with the weaker ones among the CS/PEO chains [36]. The fiber imination triggered a progressive elevation of the decomposition maximum along the imination degree, from 297 °C to 316 °C. Moreover, the chair residue at 600 °C increased when the content of imine units in chitosan fibers increased. The higher energy necessary for the fiber decomposition was explained by the stronger intermolecular forces developed by imine units, their intensity increasing along the imination degree. It can be envisaged that the conversion of amine groups of chitosan into imine units reinforced the fibers by additional intermolecular forces.

### 3.3. Swelling Behaviour

The studied fibers are intended for wound healing applications. In this regard, it is important to know their ability to swell, allowing exudate drainage while maintaining a moist environment, especially in the first days, which are critical for wound healing [37]. The fiber swelling was assessed by measuring the mass equilibrium swelling (MES) in water and PBS of different pH values (Figure 7).

The fibers displayed rapid swelling when in contact with water, attaining an MES of approx. 30 in less than 1 h (Figure 7a). No significant differences were noticed for the imino-chitosan fibers versus neat chitosan ones, demonstrating that the swelling ability was less affected by functionalization. Compared to other MES values reported in the literature for chitosan-based nanofibers, this swelling is remarkable [38], and it was attributed to the microporous nature and good hydrophilicity, as established by vapor sorption measurements. No significant MES changes were observed in the medium of physiological pH (pH = 7.4), but a progressive diminishing to 20 was noticed along with the pH increasing to 10 (Figure 7b). This was in agreement with the chitosan deprotonation in an alkaline medium [39]. On the contrary, in an acidic pH of 5.5, the fibers started to decompose, making the MES measurements impossible.

### 3.4. Fiber Mat Biodegradation

To overcome the drawback of traumatic debridement of traditional wound dressing products, it is desirable to develop bio-absorbable materials, which slowly biodegrade during the healing process. Clinical investigations revealed that the pH of wound surface and exudate is a dynamic biomarker characteristic for different healing stages. It increases from 8.5 in the first minutes to 10 in the first four days and then slowly decreases to the normal pH of the dermis (5.5) during the re-epithelization process until day 14 [40]. Also, a different level of lysozyme for non-infected wounds (376 ± 240 U/mL) compared to infected ones (4830 ± 1848 U/mL) was demonstrated [41]. In light of these outcomes, the fibers’ biodegradation was investigated in lysozyme media (376 U/mL) of different pH: 5.5, 7.4, 8.5, 9 and 10, corresponding to different healing stages (Figure 8 and Appendix A). The influence of the higher lysozyme concentration (4830 U/mL) was investigated too.

In a medium having a physiological pH (pH = 7.4), the biodegradation rate increased once the imination degree of fibers increased, reaching a mass loss of 28% for **BC1** compared to 17% for neat **CS**, after 21 days (Figure 8a). The biodegradation rapidly advanced during the first three days and continued then in a slower manner, according to the degradation mechanism of chitosan in lysozyme, consisting in the hydrolysis of O–C bonds between alternative N-acetyl units followed by dissolution of the resulting oligomers [42]. The higher degradation rate of fibers containing a higher amount of imine units was ascribed to the shifting of the imination equilibrium to the reagents in the wet media (Appendix A). Considering that the low acetylation degree of chitosan is unfavorable for enzymatic biodegradation [42], it is very probable that the biodegradation rate was influenced by the microporous nature of fibers.

Regarding the influence of pH on biodegradation rate, it was observed that the pH increase was accompanied by a mass loss increase, reaching in 21 days almost 33% for the pH = 8.5, characteristic of the first healing stage and 40% for the pH = 9, characteristic of the second healing stage (Figure 8b). In the medium of pH = 5.5, characteristic of a normal dermis, the samples completely degraded in 3 days.

The incubation of the samples in a lysosome medium of concentration characteristic of infected wounds (4830 U/mL) showed a slow increase in mass loss compared to that characteristic to non-infected wounds (376 U/mL), indicating that biodegradation was a little sensitive to the enzyme concentration (Appendix A).

Fitting the biodegradation data on the scenario of wound healing, it can be envisaged that in the first healing stage when the pH is 8.5, the fibers start to biodegrade releasing the bioactive aldehyde, the biodegradation intensifies during the second healing stage when the pH increases to 10, and the fibers decompose during the third healing stage when the pH turns to 5.5, that of the normal dermis. This indicates that the microporous chitosan fibers should progressively decompose over the wound healing period to have good chances, as traumatic debridement is to be avoided.

### 3.5. Antimicrobial Activity

It was demonstrated that invasive infections are the leading causes of death after burn injury [2], particularly those induced by bacteria able to rapidly develop biofilms (e.g., *S. aureus*), which favor the alkaline environment of wounds [43]. Candida species are also hazardous pathogens for wound injuries because they easily penetrate into deeper tissues and the bloodstream, threatening the patient’s life [44].

Considering these clinical data, the antimicrobial activity of the studied fiber mats was tested on representative Gram-positive (*S. aureus*) and Gram-negative (*E. coli*) bacteria and representative yeast (*C. albicans)* and fungal (*A. brasiliensis)* strains. Even if chitosan is known for its antimicrobial activity [45], the neat **CS** fibers did not inhibit the microbial growth around them, while inhibition zones appeared around the imino-chitosan fibers (**BC1**–**BC6**) against *C. albicans* (Figure 9b), *S. aureus* (Figure 9c) and *A. brasiliensis* (Figure 9d). The fiber activity increased along the imination degree, but strong activity was noticed even for the **BC6** sample with the lowest content of imine units (Figure 9a). The high difference in the imination degrees of the samples did not appear to be reflected in significant differences in the antimicrobial activity, suggesting that it was not correlated with the imine amount but with imination reversibility [21,29,46]. It is expected that, in moist microbial media, the imination equilibrium would shift to the reagents, that is, the antimicrobial 2-formylphenylboronic acid, as the UV-vis measurements indicated (Appendix A) [47]. Once the antimicrobial aldehyde is consumed in the inactivation process of bacteria/fungal strains, the imination equilibrium is further shifted to reagents, releasing new aldehyde amounts. Thus, the release of antimicrobial aldehyde occurs “on demand” as long as the pathogens exist in the media. These findings are of particular interest for burn healing, for which *S. aureus* is responsible for 69.5% of infections [48], and the *Candida* and *Aspergillus* species severely lead to high mortality rates in patients with major burns [49].

A visual insight by SEM microscopy on the samples after being in contact with bacteria/fungal strains exhibited rare *S. aureus* cells and no biofilm formation. An anastomosing of the functionalized chitosan fibers into the microbial media was remarked, up to form a net that entrapped the bacterial cells (Figure 10a,b). In the case of *C. albicans*, the fibers became tighter anastomosing with the yeast cells and did not allow the biofilm formation (Figure 10c). Similar observations were also valid for the samples treated with *A. brasiliensis*, when solitary spores trapped into the fiber mats were spotted (Figure 10d). No such effect was detected for the neat chitosan fibers, highlighting the antimicrobial activity of 2-formylphenylboronic acid [47], which was slowly released in the microbiologic media. It was concluded that the functionalization of chitosan fibers creates a barrier to bacterial/fungal infections.

### 3.6. Biocompatibility

Keeping in mind the application of the fibers for wound healing, their biocompatibility was investigated when in contact with normal human dermal fibroblasts (NHDF) for 24 h, by MTS assay, when reduction of cell viability more than 30% was attributed to a cytotoxic effect, according to ISO 10993-5:2009 (E) for biologic evaluation of medical devices [50]. The results indicated that except for the **BC1** and **BC1R**, the fiber mats meet the criterion of biosafety assessment (Figure 11a). The cytotoxicity of the **BC1** and **BC1R** samples was asserted to the high content of imine units at the fiber surface, and it is expected to decrease once the antimicrobial aldehyde is consumed during the healing process. The examination of the fibroblasts in contact with the fibers revealed that indeed **BC1**, **BC1R** and even **BC2** samples inflicted the modification of the cell morphology from a spindle shape to a round one, indicating cell death. No such effect was discriminated for **BC4** and **BC6** samples (Figure 11b–d and Appendix A). It was concluded that the fibers were completely lacking in toxicity for a conversion degree of the amino groups of chitosan into imine units less than 20%.

Further, the analysis of the fibers after being in contact with fibroblasts showed normal, elongated cells on the **BC6** sample and round ones for the others, indicating that a 16% imination degree will allow the cell attachment and proliferation (Figure 11e–g). In the scenario of aldehyde consumption during the healing process, it is envisaged that after 24 h, the fibroblasts can attach and proliferate on the fiber mats, restoring the tissue.

## 4. Conclusions

A series of microporous chitosan nanofibers functionalized via imine linkage with 2-formylphenylboronic acid were prepared and characterized with the aim to develop a new biomaterial design for wound dressings. Microporous chitosan nanofibers, with average diameters lower than 200 nm, inter-fiber pores around 1 µm and intra-fiber micropores up to 2 nm, were successfully reached by electrospinning of a chitosan/PEO blend followed by PEO removal. The functionalization of the fibers by imination in a heterogeneous system with 2-formylphenylboronic acid allowed controlling the imination degree (52.8 to 14.7%) and the preponderant disposal of imine units (in the micropores or at the fiber surface) by varying the molar ratio of functional groups and the rate of solvent removal from the reaction system. The microporous nature of the fibers favored rapid swelling in water and phosphate buffer of different pH, reaching a remarkable MES around 30. The enzymatic biodegradation in media of different pH corresponding to the pH evolution over the healing period of wounds, showed an increase of biodegradation rate along with a pH increase to 10, corresponding to the first two stages of wound healing, and the rapid biodegradation in medium of pH 5.5 characteristic for the normal dermis. The functionalization with 2-formylphenylboronic acid endowed the fibers with antimicrobial activity against opportunistic pathogens such as *S. aureus*, *A. brasiliensis*, and *C. albicans* even for the lowest imination degree (14.75%), correlated with the shifting of imination equilibrium to the reagents in wet media. Biocompatibility tests on normal human dermal fibroblasts according to ISO 10993-5:2009 (E) indicated the safe use of the functionalized fibers in contact with cells up to 20% imination degree, while an imination degree up to 16% prompted the cell adhesion. Cumulating these findings, it can be summarized that the preparation of microporous chitosan nanofibers and their condensation with 2-formylphenylboronic acid to reach a 16% imination degree of chitosan led to biomaterials with (i) ability to swell and thus to drain the wound exudate favoring a moist medium proper for healing (ii) biocompatibility and bioadhesivity beneficial for tissue repair; (iii) antimicrobial activity beneficial for preventing the wound infection in the first stages of healing (iv) biodegradation rate fitting well the wound healing period, pointing for avoiding of traumatic debridement. All these cumulative properties satisfying the requirements of an ideal wound dressing encourage further investigations in order to provide a marketable product. Moreover, the paper brings to researchers’ attention a new promising design for wound dressings, consisting of microporous chitosan nanofibers functionalized by imination with antimicrobial agents.

## Data Availability

Data supporting the results were provided in the Appendix A.

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
