# Peer review of "Imination of Microporous Chitosan Fibers—A Route to Biomaterials with “On Demand” Antimicrobial Activity and Biodegradation for Wound Dressings"

_pharmaceutics, 2022, doi:10.3390/pharmaceutics14010117_

Round 1
Reviewer 1 Report
The manuscript presents a chitosan based nanofibrous wound dressing that poses "on demand" features like biodegradation of polymer and antimicrobial properties. The chitosan fiberes were iminated with 2-formylphenylboronic acid to make the material anti-bacterial. The manuscript preents all the stages of biomaterial fabrication with many characterization steps. Specific comments: Page 2 line 59-62 Chitosan is bioabsorbable polymer, but what is important is the type of the biomaterial that can be in the e.g. hydrogel form or electrospun mat. In the first case it is replaced after 1 to 3 days with a new dressing. In case of hydrogels the adhesion is low, but as far I understand, your dressing would stay on the wound? Page 2 line 72-77 Synthetic polymers can be biodegradable. Page 2 line 81line Why is the information of pore size of the order of 2 nm important? Page 2 line 84-85 Cam you control externaly the release of antimicrobial agent? In my opinion you cannot, therefore it is not on demand release. Page 8 Figure 1. It is not clear what you show in image a). Please zoom in for specific wavenumber range and put wide range into supporting info. Page 10 Figure 3. What is the reason to pleace image h). It is not informative. Line 373 Ordering can be observed with SEM. Line 378. Can you use other gas sorption technique to differentiante the surface area per g of the mat for samples CS/PEO and CS? Page 13 Figure 5. MES should be presented in %, and instead of wrtiting MES you should write Water absorption simply. The ability to absorb water is tremendous. Page 14 Figure 6a and b. Where are standard deviations and statistical analysis? And why do you show BC1R samples in figure b) when you know that this biomaterial is cytotoxic? Line 502-503 You did not perform in vivo study and therefore you can not claim that there is no need for traumatic debridment. I don't see to many references to papers where chitosan nanofibers were analysed. Line 505 Probably the principal cause of burn death is fire. Please rewrite this sentence or provide reference to statistics. Line 523 Can you please explore more on this imination reversability? Can you please perform the drug release analysis? Figure 8 c) and d) I don't understand what I see on those images. PLease provide more information in caption. Figure 9. The e)-g) images are of poor quality showing just few cells. Do you have images after 3 or 7 days of culture? Line 593 2 nm pore is micro sized?Author Response
Answer to Reviewer 1
Reviewer comment
The manuscript presents a chitosan based nanofibrous wound dressing that poses "on demand" features like biodegradation of polymer and antimicrobial properties. The chitosan fiberes were iminated with 2-formylphenylboronic acid to make the material anti-bacterial. The manuscript preents all the stages of biomaterial fabrication with many characterization steps. Specific comments: Page 2 line 59-62 Chitosan is bioabsorbable polymer, but what is important is the type of the biomaterial that can be in the e.g. hydrogel form or electrospun mat. In the first case it is replaced after 1 to 3 days with a new dressing. In case of hydrogels the adhesion is low, but as far I understand, your dressing would stay on the wound?
Author answer
Tests realized on materials mimicking the tissues (paper swelled in buffer solution of physiologic pH 7.4) showed good adhesivity of these fibers, possible due to their rapid swelling favored by the inter- and intra-fiber pores. Also, the fiber mats easily adhered to glass, metal or leaves when they were moisture. These are good premises that the fibers will stay on wounds. Images were given in Supporting Information and the text has been updated to indicate this.
Reviewer comment
Page 2 line 72-77 Synthetic polymers can be biodegradable.
Author answer
We agree with the reviewer that there are synthetic polymers which can biodegrade. However, polyethylene glycol and polyvinyl alcohol, which showed the best ability to act as chitosan co-spinning agents, are not in vivo biodegradable. The text has been modified in order to reflect this and proper references were cited.
Reviewer comment
Page 2 line 81line Why is the information of pore size of the order of 2 nm important?
Author answer
We highlighted the size of pores, in order to be clearer in attributing the terminology of “microporous”, to not be confused with the existence of inter-fiber pores which are around one micrometer. As you know, according to IUPAC, the microporous materials refer to materials with pores lower of 2 nm.
Reviewer comment
Page 2 line 84-85 Cam you control externaly the release of antimicrobial agent? In my opinion you cannot, therefore it is not on demand release.
Author answer
The imine reversibility test suggested that the release of the antimicrobial aldehyde will be controlled by its consumption in the process of fungi/bacteria growth inhibition. This means that the occurrence of an infection will determine a higher consumption of aldehyde and consequently a higher release. In our opinion this can be considered “on demand”. To better highlight this, we realized an in vitro release experiment, which clearly show that the aldehyde release is controlled by its consumption. The text has been modified to highlight this.
Reviewer comment
Page 8 Figure 1. It is not clear what you show in image a). Please zoom in for specific wavenumber range and put wide range into supporting info.
Author answer
The Figure 1a) was magnified for better quality. Besides, the Figure has been given in Supporting Information too in the original format for clarity.
Reviewer comment
Page 10 Figure 3. What is the reason to pleace image h). It is not informative. Line 373 Ordering can be observed with SEM.
Author answer
Image h) shows the birefringence of the fibers under polarized light, indicating that the chitosan chains were aligned during the electrospinning. This gives information about the fiber crystallinity. The text has been modified to better reflect this. The image was replaced with one of better quality.
Reviewer comment
Line 378. Can you use other gas sorption technique to differentiante the surface area per g of the mat for samples CS/PEO and CS?
Author answer
Vapor water sorption technique has been chosen to evidence the intra-fiber pores of the studied samples and also to evidence their hydrophilicity. Unfortunately, in a such short period of time we can’t manage to perform another gas sorption technique. Neverthless, this technique offered the information required in order to understand the behavior of the studied materials.
Reviewer comment
Page 13 Figure 5. MES should be presented in %, and instead of wrtiting MES you should write Water absorption simply. The ability to absorb water is tremendous.
Author answer
As you know, in the literature some authors asses the swelling by calculating the mass equilibrium swelling (MES) or water absorption. Both variants are correct and largely used. Up to now, in our studies we used MES, and for comparison reasons we prefer to use it further. This parameter appears particularly suitable for these materials, as the equilibrium swelling was established very fast, in less than 1 hour. This was the most probably correlated to the microporous nature of the fibers, which strongly impacted their capacity do adsorb liquids reaching a MES up to approx. 30. The text has been changed to better reflect this.
Reviewer comment
Page 14 Figure 6a and b. Where are standard deviations and statistical analysis? And why do you show BC1R samples in figure b) when you know that this biomaterial is cytotoxic?
Author answer
We apologize for this oversight. The tests were done in triplicate, and the average values were used for graphical representation. The graphs were re-drawn to comprise the standard deviations, in Figure 6 and Figure 5 as well. At the moment when these tests were done, we didn’t know the fiber cytotoxicity. We have chosen these fibers because their enzymatic biodegradation at pH 7.4 was quite similar to the other samples. At this moment we started another biodegradation experiment on the BC4 sample, hoping we will can include the results in the final version of the paper (the experiment duration is 21 days, more than 10 days we had for paper revision).
Reviewer comment
Line 502-503 You did not perform in vivo study and therefore you can not claim that there is no need for traumatic debridment.
Author answer
The reviewer is right, no in vivo tests were done at this stage. We envisaged the potential of avoiding the traumatic debridement due to the good match between the biodegradation rate in media with pH corresponding to that of exudate over the healing period. This was better highlighted in the text.
Reviewer comment
I don't see to many references to papers where chitosan nanofibers were analysed.
Author answer
As can be seen, the reference 14 is a review on chitosan nanofibers, comprising more than 200 references focusing on chitosan nanofibers, published in the last 20 years. Moreover, the reference 5 is another review focused on polysaccharide fibers for wound management. Other 5 papers focused on chitosan nanofibers for wound healing were cited as well. But the reviewer is right, their percent in the total of cited references is low. In this light, along the paper we have cited other specific papers.
Reviewer comment
Line 505 Probably the principal cause of burn death is fire. Please rewrite this sentence or provide reference to statistics.
Author answer
Actually, many statistics showed that the leading cause of death after burn injury is the invasive infection. This was highlighted also in Introduction with a proper reference (see ref. 2). The text has been modified to better highlight this.
Reviewer comment
Line 523 Can you please explore more on this imination reversability? Can you please perform the drug release analysis?
Author answer
The imination reversibility has been explored by UV-vis, in static and dynamic conditions. At the reviewer suggestion, we performed the investigation of aldehyde release, and the manuscript has been updated consequently.
Reviewer comment
Figure 8 c) and d) I don't understand what I see on those images. PLease provide more information in caption.
Author answer
The caption was rewritten to indicate that not biofilm formation was observed on the fiber samples.
Reviewer comment
Figure 9. The e)-g) images are of poor quality showing just few cells. Do you have images after 3 or 7 days of culture?
Author answer
The test was done by incubation of the cells within 24 hours, according to a literature protocol (ref. 24). The images show that after 24 hours, the cell attachment started. On the images of lower magnification, the cells are not noticeable confusing with the fibers, this is why we have chosen the images which magnification allowed to see the fibers and cells as well. Unfortunately, in the short period of time given for manuscript revision and in this period of the year (winter celebrations when many researchers are in holyday), we could not manage to perform other measurements at 3 or 7 days of incubation.
Reviewer comment
Line 593 2 nm pore is micro sized?
Author answer
We know that this terminology can be confusing, but according to IUPAC terminology, pores with diameter lower than 2 nm are called micropores (see. Reference 32).
We thank to the reviewer for the professional feed-back which helped us to see the weaknesses of our paper and to improve it for the readers benefit. All changes in the manuscript were highlighted in red letters.
Yours sincerely,
Luminita Marin

Reviewer 2 Report
The manuscript analyzes a current and important topic, is well structured and presents the results clearly. I am of the opinion that the article should be published.
Author Response
Answer to Reviewer 2
Reviewer comment:
The manuscript analyzes a current and important topic, is well structured and presents the results clearly. I am of the opinion that the article should be published.
Author answer:
We thank to the reviewer for her/his positive feed-back. Your expertize and kindness to spend time to read our article is highly appreciated.

Reviewer 3 Report
General comment:
The paper presents a study of the material for the wound healing process. This material is based on Chitosan, which is coated with antimicrobial agents via imine linkage. The study shows the promising benefit of the “on demand” antimicrobial properties and biodegradation rate matching the healing stages. My main concern is as follows:
Comment 1:
There are a lot of studies in literature regarding use of Chitosan for the same purpose, see the literature review paper “A Review of the Antimicrobial Activity of Chitosan” https://www.scielo.br/j/po/a/LKwBpFnrWSJ3JwFXzwMNpgF/?format=pdf&lang=en. There are also a lot of studies of using Chtosan to make hydrogels to serve as a scaffold or encapsulates, see the recent work “Synthetic Artificial Pancreas Using Chitosan Hydrogels Integrated with Glucose-Responsive Microspheres for Insulin Delivery”.
Comment 2:
Natural biomaterial, Chitosan, is of course good in terms of biocompatibility, but it is shortcoming, e.g., weak strength, etc. As such, in literature synthetic materials are used for wound healing dressing, e.g., “Electrospun PLGA membrane incorporated with andrographolide-loaded mesoporous silica nanoparticles for sustained antibacterial wound dressing”. The authors may need to comment on both approaches. The present version of the manuscript seems missing such a comment notwithstanding comparison.
Comment 3:
Regarding the “on-demand” biodegradation, there is a work called active control of degradation, see the review paper “Control of Scaffold Degradation in Tissue Engineering: A Review. Tissue Engineering B”. The authors may want to comment on whether the active degradation control makes sense to their application. I feel that the current approach to “on demand” is passive and its accuracy may not meet the requirement.
Comment 4:
There are some English writing errors, but they can be easily fixed. My main concern is the novelty and originality of the work.
Author Response
Answer to reviewer 3
Reviewer comment:
General comment:
The paper presents a study of the material for the wound healing process. This material is based on Chitosan, which is coated with antimicrobial agents via imine linkage. The study shows the promising benefit of the “on demand” antimicrobial properties and biodegradation rate matching the healing stages. My main concern is as follows:
Comment 1:
There are a lot of studies in literature regarding use of Chitosan for the same purpose, see the literature review paper “A Review of the Antimicrobial Activity of Chitosan” https://www.scielo.br/j/po/a/LKwBpFnrWSJ3JwFXzwMNpgF/?format=pdf&lang=en. There are also a lot of studies of using Chitosan to make hydrogels to serve as a scaffold or encapsulates, see the recent work “Synthetic Artificial Pancreas Using Chitosan Hydrogels Integrated with Glucose-Responsive Microspheres for Insulin Delivery”.
Author answer
We thank to the reviewer for the interesting articles indicated. We read them and those related to our paper were considered.
Reviewer comment:
Comment 2:
Natural biomaterial, Chitosan, is of course good in terms of biocompatibility, but it is shortcoming, e.g., weak strength, etc. As such, in literature synthetic materials are used for wound healing dressing, e.g., “Electrospun PLGA membrane incorporated with andrographolide-loaded mesoporous silica nanoparticles for sustained antibacterial wound dressing”. The authors may need to comment on both approaches. The present version of the manuscript seems missing such a comment notwithstanding comparison.
Author answer:
We agree with the reviewer that literature data indicate weak strength of the chitosan-based materials. However, as can be seen in the figure S6, the prepared chitosan-based nanofibers have good integrity and can be easily manipulated, moreover, a mat of 10 mg sample was able to sustain a weight of 50 grams (manuscript updated with images in Supporting Information). Furthermore, they proved biodegradability, with a biodegradation rate depending on pH, which suggested that they should be completely biodegraded during the healing period of wounds. The text has been modified to highlight this.
Reviewer comment:
Comment 3:
Regarding the “on-demand” biodegradation, there is a work called active control of degradation, see the review paper “Control of Scaffold Degradation in Tissue Engineering: A Review. Tissue Engineering B”. The authors may want to comment on whether the active degradation control makes sense to their application. I feel that the current approach to “on demand” is passive and its accuracy may not meet the requirement.
Authors answer
We have read the indicated review paper. The meaning of “on demand” in our paper is certainly different compared to that of “active degradation control”. In our paper we didn’t refer to the fact that the degradation rate is externally monitored, but to the fact that, the biodegradation rate of the prepared materials is controlled by the pH of the medium (i.e. exudate pH over the wound healing period). The aim of the paper was to design materials as bioabsorbable bandages for wound healing. The biodegradation study in media mimicking the wound exudate over the wound healing period indicates that the materials degradation is controlled by the wound pH. We didn’t find such a study of enzymatic degradation of chitosan in literature, even in the indicated review paper.
Reviewer comment:
Comment 4:
There are some English writing errors, but they can be easily fixed. My main concern is the novelty and originality of the work.
Author answer:
Recently, we reviewed the findings in the chitosan-based nanofibers domain, published in the last 20 years (more than 200 articles, see the reference 16). Among these, we couldn’t find any paper dedicated to the preparation of microporous chitosan fibers for wound dressings, and not at all the idea of bonding an antimicrobial agent by reversible covalent bonds which favors the antimicrobial agent release by shifting the imination equilibrium to the reagents under the external stimuli (consumption). We think these are original approaches for wound healing materials. Besides, the design of these materials was confirmed by good findings: fast swelling, biodegradation rate controlled by pH of exudate reaching total biodegradation at the pH of dermis, antimicrobial properties and biosafety.
We want to thank to the reviewer for the constructive feed-back on our paper, which helped us to improve it for the readers benefit. The changes along the manuscript were highlighted in red letters.
Yours sincerely,
Luminita Marin

Round 2
Reviewer 1 Report
The authors improved the manuscript, however, I'm still not convinced with several explanations e.g. "on-demand" drug delivery. Nevertheless, it can be considered for publication.
Reviewer 3 Report
I am satisfied with the revision and rebuttal.